# Improving the Corrosion Protection of Poly(phenylene methylene) Coatings by Side Chain Engineering: The Case of Methoxy-Substituted Copolymers

**DOI:** 10.3390/ijms232416103

**Published:** 2022-12-17

**Authors:** Marco F. D’Elia, Mirko Magni, Stefano P. M. Trasatti, Markus Niederberger, Walter R. Caseri

**Affiliations:** 1Laboratory for Multifunctional Materials, Department of Materials, ETH Zürich, 8093 Zürich, Switzerland; 2Department of Environmental Science and Policy, Universitá degli Studi di Milano, 20133 Milan, Italy

**Keywords:** poly(phenylene methylene), thermoplastic coating, electrochemical impedance spectroscopy, self-healing, side-chain engineering

## Abstract

This work aims to improve the corrosion protection features of poly(phenylene methylene) (PPM) by sidechain engineering inserting methoxy units along the polymer backbone. The influence of side methoxy groups at different concentrations (4.6% mol/mol and 9% mol/mol) on the final polymer properties was investigated by structural and thermal characterization of the resulting copolymers: co-PPM 4.6% and co-PPM 9%, respectively. Then, coatings were processed by hot pressing the polymers powder on aluminum alloy AA2024 and corrosion protection properties were evaluated exposing samples to a 3.5% w/v NaCl aqueous solution. Anodic polarization tests evidenced the enhanced corrosion protection ability (i.e., lower current density) by increasing the percentage of the co-monomer. Coatings made with co-PPM 9% showed the best protection performance with respect to both PPM blend and PPM co-polymers reported so far. Electrochemical response of aluminum alloy coated with co-PPM 9% was monitored over time under two “artificially-aged” conditions, that are: (i) a pristine coating subjected to potentiostatic anodic polarization cycles, and (ii) an artificially damaged coating at resting condition. The first scenario points to accelerating the corrosion process, the second one models damage of the coating potentially occurring either due to natural deterioration or due to any accidental scratching of the polymer layer. In both cases, an intrinsic self-healing phenomenon was indirectly argued by the time evolution of the impedance and of the current density of the coated systems. The degree of restoring to the “factory conditions” by co-polymer coatings after self-healing events is eventually discussed.

## 1. Introduction

Aluminum alloy AA2024 is one of the most used alloys in aerospace industry due to its favorable strength to weight ratio [1]. However, due to the heterogeneous microstructures present in the alloy matrix, AA2024 is particularly susceptible to localized corrosion especially if chloride ions are present even at low concentration [1,2,3,4,5]. In the metal industry, the corrosion protection strategy adopted to protect this aluminum-based alloy comprises the application of chromium (VI) in many steps of manufacturing [1,6,7]. Currently, the replacement of chromium-based compounds is one of the most important challenges in order to reduce the hazards and risks within these products [8,9].

Among the various strategies to prevent corrosion, poly(phenylene methylene) (PPM) has been recently found to be a valuable material to obtain smart anticorrosion organic coatings for aerospace alloys [10,11,12,13]. Although the polymer presents a set of valuable properties for protective coatings, such as a high thermal stability [13,14], hydrophobicity [14,15], resistance toward oxidation [16], good barrier properties [14,17], and UV-triggered fluorescence [18], brittleness has significantly hampered its exploitation. Only recently the processability problems were overcome allowing to craft films free of cracks. The first strategy to obtain homogeneous and uniform coatings reports the use of blends containing PPM formulated with an external plasticizer (i.e., benzyl butyl phthalate) [12]. A plasticizer causes a decrease of the polymer–polymer interactions resulting in a softer material with lower glass transition temperature and improved film-forming properties [10,19,20]. Anodic polarization tests performed on such coatings showed good corrosion protection and inhibition of corrosion propagation due to the self-healing of the coating during the pitting attack [12,21]. However, external plasticizers are often low molar mass compounds [21] which are subjected to migration phenomena leading, potentially, to an early coating degradation [22]. Therefore, in order to replace the plasticizer while preserving the polymer processability, PPM copolymers containing octyloxy side chains along the polymer backbone were synthesized [10,23]. The side chains are supposed to decrease the interactions between the polymer backbones, ameliorating their sliding properties and providing soft materials without the disadvantage of external plasticizers [10,24,25]. The octyloxy side groups not only decrease the glass transition temperature to ca. 35 °C (compared with ca. 63 °C of PPM itself) but also provide a reversible thermoplastic material which is able to recover its structure after several thermal shocks [10,11]. Due to these improvements, in addition to the increasing of the corrosion protection ability with respect to the reference blend PPM, the copolymer coatings are able to self-heal, maybe by exploiting the heat flow released during the corrosion of aluminum [11,26]. Although PPM copolymers containing octyloxy side chains show excellent performance at room temperature, the low glass transition temperature leads to a loosening of the protection ability just above room temperature [10].

Intermolecular interactions within a polymer matrix have an impact on fundamental properties for coating materials such as glass transition temperature and mechanical stability of the coating [24,27,28,29]. Polymer side-chain engineering allows the modulation of intermolecular interactions between polymer backbones, influencing van der Waals forces, material packing levels, and thermal transition temperatures [30,31,32]. In turn, tuning thermal transition temperatures is crucial in controlling relevant features of materials such as thermal stability, storage and loss modulus, and deformability of the polymer allowing to achieve high-quality coatings [24,33,34,35,36]. The side chain length has been shown to play a key role in tuning intermolecular interactions, the latter increasing as the side-chain length decreases [28,35,37]. Moreover, increasing intermolecular interactions can ameliorate several material properties such as the polymer backbone rigidity, polymer chains orientation, and self-healing which are fundamental key factors for valuable anticorrosion coatings [28,29,38,39,40].

In the present study, we designed PPM copolymer coatings containing methoxy groups. Their corrosion protection performance was assessed for AA2024 aluminum alloy. The replacement of long chains (octyloxy pendants) with shorter ones (methoxy groups) is aimed at increasing the glass transition temperature but maintaining the film-forming properties of substituted PPM derivatives [24,27]. For this purpose, *p*-methoxy benzyl chloride was used as comonomer to benzyl chloride to synthesize a PPM derivative endowed with short methoxy side chains distributed along the polymer backbone. Gratifyingly, methoxy-PPM derivatives showed also an improved corrosion protection ability with respect to similar coatings reported so far [10,11,12]. Moreover, a specific investigation protocol was developed to prove the corrosion triggered self-healing mechanism of the methoxy-PPM-based coatings.

## 2. Results and Discussions

### 2.1. Polymer Side Chain Engineering and Characterization of the Obtained Copolymers

As previously reported [18,41], PPM can be synthesized by polymerization of the monomer benzyl chloride and PPM copolymers with side chains with corresponding benzyl chloride derivatives as comonomers. Thus, in the synthetic route applied here, the commercially available *p*-methoxybenzyl chloride was added as comonomer to benzyl chloride to yield two products with 4.6% mol/mol (co-PPM 4.6%) and 9.0% mol/mol (co-PPM 9.0%) methoxybenzyl units, as evaluated by integration of ^1^H-NMR spectra (see Appendix A). Besides the signals of the methoxy group between 2.8 and 3.2 ppm, the ^1^H- NMR spectra revealed broad signals in the typical range for PPM [41]. Moreover, the ^13^C-NMR spectra (see Appendix A) of the obtained copolymers showed the typical signals of PPM, in addition to the signal of the methoxy group at 68 ppm [41], i.e., there was no evidence for a change in the substitution patterns with respect to PPM itself [41]. The weight average molecular weights and polydispersity indexes (PDI) of the copolymers containing 4.6% mol/mol and 9.0% mol/mol amounted to 9997 g/mol (PDI 3.26) and 12,125 g/mol (PDI 4.47), respectively. The copolymers were soluble in solvents used for pristine PPM (e.g., THF and chlorinated solvents).

Thermogravimetric analysis (TGA) confirmed the high thermal stability of the materials which virtually did not lose mass below 400 °C. Differential scanning calorimetry (DSC) revealed a glass transition temperature (T_g_) of the copolymers of about 65 °C (Appendix A), achieving the expected effect by the introduction of short aliphatic chains along the polymer backbone. For comparison, (i) the octyloxy-containing PPM copolymers previously published exhibited T_g_ between 31 °C and 48 °C, depending on the molar percentage of the comonomer [10], and (ii) the PPM homopolymer exhibited T_g_ around 60° [42]. The lack of any other thermal transition for the methoxy-bearing PPM copolymers indicates complete amorphous behavior, as already reported for PPM and derivatives [10,18]. The insertion of methoxy side chains did not affect the UV–Vis absorption spectrum of the co-polymer that is consistent with the electronic transitions of the homopolymer PPM [18]. Similarly, the photoluminescence spectra of the copolymer still present two emission phases (called blue and green phase, occurring under an excitation of 386 and 455 nm, respectively), as previously reported for pure PPM (Appendix A) [18].

### 2.2. Corrosion Protection of AA2024 by PPM Copolymers Coatings

Figure 1 compares the potentiodynamic anodic polarization curves for a pristine AA2024 surface and for same alloy specimens coated with a 30 µm layer of co-PPM 4.6% and co-PPM 9.0% exposed to a 3.5% w/v NaCl solution. The potentiodynamic polarization is a useful technique to evaluate the insulating and protective action of organic coatings over conductive substrates. Additionally, in the case of exposure of a metal to a sodium chloride solution, the technique allows also to evaluate the barrier properties against the diffusion and migration within the coating layer of chlorides, which are the anion species mainly responsible for localized corrosion events of passive metals [5,42,43]. The anodic curves of the samples are strongly affected by the covering of the surface with the copolymer. In the case of naked AA2024, an abrupt increase of the current density, from around 0.1 µA cm^−2^ to some mA cm^−2^, is recorded at very low overpotentials (i.e., at potentials slightly more positive than the free corrosion potential). This trend reflects the transpassivity behavior of the metal alloy expected in presence of Cl^−^ anions that makes hardly reaching the protection condition during the cathodic backward scan (around 300 mV more negative than the free corrosion potential). In contrast, the deposition of the coating invariably resulted in a considerable increase of the stability window of the alloy according to a marked decrease of the current density, constantly in the order of pA cm^−2^, confirming the good protection ability (i.e., low corrosion rate) at least up to 0 V vs. SCE (around 700 mV anodic overpotential). Interestingly, a significant difference exists between the two copolymers. The co-PPM 4.6% coating exhibits an abrupt increase of the current density at higher potentials reflecting the coating failure as shown in Figure 2 (i.e., holes and delamination areas). The electrochemical data indicate that, at this lower concentration, methoxy units do not drastically improve the film-forming properties of PPM. Imperfections (voids, micro cracks, etc.) within the coatings thus lead to the permeation of water and chloride anions that trigger the localized corrosion of the aluminum and the progressive loss of the integrity of the film. These events correspond experimentally to the progressive increase of the current up to an almost stable value attributable to diffusion limitations of reagents/products in the solution. On the other hand, co-PPM 9% coating revealed a very low and stable current density on the order of pA cm^−2^ [18] within the whole range of the investigated potential, that is more than 3 V vs. OCP. This indicates that the higher amount of comonomer assures a compact and almost defect-free polymer matrix which prevents the diffusion of water and the migration of chloride anions from the solution to the underneath aluminum surface [18,28,29,35,37,38,39,40].

Consequently, the physical barrier action of co-PPM 9% coating provided an efficient protection of the metallic surface which is prevented from triggering any localized corrosion that in turn helps in maintaining almost unalerted the coating (Figure 2). The uniformly distributed current spikes of the polarization curves probably derive from the change of chloride concentration through the micro porosities of the coating which is not enough to trigger corrosion phenomena but leads to small metastable pitting events [11,44].

Co-PPM 9% coating exhibited the best performance (i.e., the lower current density under a potentiodynamic anodic polarization, see Appendix A) with respect to analogue PPM-based coatings, blend or co-polymer ones, already reported in literature [18]. The improved barrier effect with respect to the octyloxy PPM co-polymer can be attributed to the stronger intermolecular interactions of the polymer chains as a result of the shorter length of the methoxy pendants [28,35,37]. As confirmed by reproducibility tests (Appendix A), the better performance of co-PPM 9% coatings led us to pursue further investigations on this copolymer.

### 2.3. Behavior of the Coating under a Stress Condition via Anodic Ageing

Spikes in the anodic polarization curves of coated specimens suggested the formation of meta-stable pits (short lasting local corrosion processes) that could be attributed to microscale self-healing events as already reported for analogues PPM-based coatings [2,3,4,5,11,43] Thus, in order to assess the long-time effect of such corrosion processes on the overall behavior of the coated system, co-PPM 9.0% coated samples were subjected to an accelerated aging process. Through a multi-step cycling procedure, the sample is firstly anodically polarized for 10 min at 0 V vs. SCE (i.e., ca. 700 mV anodic overpotential) and, after a relaxation period to the equilibrium potential, it is investigated by electrochemical impedance spectroscopy (EIS).

The effect of the accelerated anodic aging is reported in the Bode modulus plots (Figure 3 left). Even after 6 anodic polarization cycles (cumulative time = 1 h), the coating exhibited an effective barrier property towards the solution. The Bode spectra show a pure capacitive behavior, well modeled by an equivalent electric circuit that comprises a resistor with a capacitor in series. Coherently, the corresponding current densities obtained over the anodic polarization cycles (Figure 3 right) are in the order of tens pA cm^−2^, whose values can be easily ascribed to a small “leakage current” fluxing across the dielectric polymer layer due to the partial permeation of the solution within the porosity of the polymer matrix.

The slight, but perceivable, increase of current density recorded during the 7th polarization step, the emergence of a breakpoint frequency at ca. 100 Hz in the impedance spectra and the decrease of the impedance modulus at lower frequencies suggest a loss of insulation from the coating that triggers corrosion reaction. The progressive increase of the current density by up to four orders of magnitude (few tens µA cm^−2^) recorded in the subsequent cycle reflects a further increase of the active area at the metal|solution interface as a result of a wider delamination/rupture area of the coating. Specularly, breakdown frequency shifts towards a higher value and |Z|_0.1Hz_ decreased due to a corrosion phenomenon involving a wider area. Interestingly, during the 9th polarization step, the electrochemical markers point to the re-establishment of an insulating system (i.e., a marked drop of the current density down to pA cm^−2^ and |Z|_0.1Hz_ around 10^10^ Ω cm^2^). The behavior cannot be ascribed only to the formation of corrosion side product due to the particular autocatalytic corrosion mechanism of aluminum-based alloys in presence of chlorides [45]. Hence, it is attributed to an intrinsic self-healing event by the PPM copolymer that diminishes surface area of the metal|solution interface where the corrosion reaction can take place. 

Despite the experimental data and the fitting parameters of EIS spectra showing a complete recovery of the insulating properties of the coating, the durability of the healed coating was not completely restored. Hence, after two polarization cycles (cycle 11), the coating failed again returning electrochemical parameters comparable to those of cycle 8 (Figure 3).

### 2.4. Behavior of the Coating after an Artificial Damage

Pristine aluminum alloy is subjected to localized corrosion even at the open circuit potential in the presence of chloride anions (Figure 1) [2,3,4,5,43]. The ability to quench this undesired event, which could occur in any area of the alloy that becomes accidentally exposed to the medium, is a fundamental feature for a protective coating. To evaluate the capability of the PPM copolymer coatings in quenching the spontaneous corrosion involving a small, exposed area of AA2024, artificially damaged co-PPM 9% coatings were studied through dynamic electrochemical impedance spectroscopy (D-EIS) under operando conditions (i.e., without perturbing the system equilibrium). The pristine coated sample was damaged making a needle hole (0.52 mm in diameter) in order to create a direct, well-confined contact area between the underneath substrate and the electrolytic 3.5% w/v NaCl solution. Periodically, EIS spectra were recorded at open circuit potential, around –0.7 V vs. SCE.

The more representative Bode modulus spectra, recorded over a period of 39 h, are reported in Figure 4. For shorter exposure times, two features prove the efficacy of the scratch in exposing the underneath AA2024 surface to the electrolyte that triggers a spatially confined corrosion:(i)the system has an impedance modulus at the lower frequencies, |Z_0.01Hz_|, slightly above 10^4^ Ω cm^2^, that is around 5–6 order of magnitude lower than that observed in a pristine one (Figure 3);(ii)an inductive loop in the Nyquist modulus spectra at the lower frequency range is observed (Appendix A). The latter is a characteristic fingerprint often observed for metals in which the pitting potential and the corrosion potential are similar (i.e., transpassivity, see curve of the naked AA2024 in Figure 1).

As the exposure time increases, the module of the impedance increased as well (Figure 4) until reaching, after around 7 h, a stable value of |Z_0.01Hz_| above 10^8^ Ω cm^2^, typical of many anticorrosion coatings in a good shape. To assess the stability of the sealed coating to a prolonged exposure, the sample was kept immersed in the solution for 2 months. The high value of the impedance modulus lasts even after 60 days of continuous exposure to the synthetic seawater solution (Appendix A). The appearance of two distinct characteristic frequencies in the spectrum also confirms the re-establishment of a good “barrier effect” in the healed area, although with a slightly lower insulating capability with respect to a fresh pristine coating (one-time constant system; see for example Figure 3).

**Figure 4 ijms-23-16103-f004:**
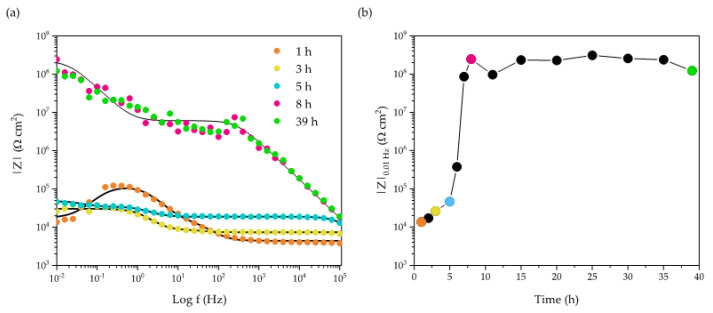
(**a**) Bode impedance spectra of a PPM copolymer coating with an artificial scratch upon exposure to 3.5% m/v NaCl solution. Dots represent experimental data and line are the fitting equations. (**b**) Evolution of |Z|_0.01Hz_ over exposure time; colors correspond to times reported in (**a**).

EIS spectra were then processed by modeling the experimental data with a proper equivalent electric circuit characterized by two-time constants (for more details, see dedicated section in Materials and Methods). Fitting parameters, as a function of exposure time, are reported in Figure 5. Coating capacitance C_c_ (Figure 5a), proportional to the content of water in the polymer matrix, slightly increases (from 10^−6^ F cm^−2^ to 10^−5^ F cm^−2^) over the first 3 h of exposure and then it drastically decreases down to 10^−10^ F cm^−2^. In a specular way, the pore resistance, R_po_, settles on values about 10^4^ Ω cm^2^ during the first 4 h of exposure and then increases up to values around 10^7^ Ω cm^2^. The trend observed for R_po_ and C_c_ in the early stage of the experiment well reflects the presence of a drilled coating with a higher surface exposed to the solution and thus prone to a higher water up-take by the polymer matrix. The drop of C_c_, and the related increase of R_po_, observed after the first hours of exposure would indicate a reversal trend that can be associated to a sudden decrease of the water up-take attributable to the healing of the original small hole by readjustment of the polymer matrix that fills the original free space [45]. The chemicals-free intrinsic self-healing of the coating is supposed to be triggered by the exothermicity of the aluminum corrosion reaction [46,47] combined with the thermoplastic behavior and the relatively low temperature of the glass transition of the PPM copolymer.

A comparable picture can be argued by looking at the behavior of the metal|solution interface, in turn strictly related with the presence or not of an insulating layer on the metal spot. The polarization resistance, R_p_, and the double layer capacitance, C_dl_, are useful to evaluate the delamination of the topcoat (i.e., the extension of the exposed area of AA2024) and the onset of the corrosion process. At the beginning, the double layer capacitance is around 10^−6^ F cm^−2^ that is a common value for an aqueous electrolyte in contact with a conductive electrode. The subsequent fast increase of C_dl_ is possibly attributable to the formation of a closed zone, of progressively smaller volume, in which the solution is entrapped between the metal surface and the polymer matrix starting to flux in the original free space. As a result of the Al corrosion occurring in a very limited volume, a fast increase of the concentration of the solution is expected and hence an increase of the dielectric constant of the solution. After around 6 h, the complete healing of the original hole leads C_dl_ to settle to values below 10^−7^ F cm^−2^, as a result of the significant reduction of real active area of Al. Coherently, while during the first hours of exposure, the polarization resistance shows values around 10^4^ Ω cm^2^, it then increases settling stably above 10^8^ Ω cm^2^, indicating a drastic decrease of the active Al surface subjected to corrosion. Hence, both R_p_ and C_dl_ confirm the switch from an active interface (i.e., corrosion phenomena) to a quenched one, with the boundary set at around 6 h.

A possible rationalization of all the phenomenological events can be provided by combining the following two statements:(i)The trend of the fitting parameters and the modification of the impedance spectra together with the increase of the impedance modulus up to 10^8^ Ω cm^2^ during time (Figure 4) are all proof of the occurrence of a coating self-healing event in the damaged area rather than the production and accumulation of corrosion products within the small spot [46,47].(ii)Aluminum alloys are particularly susceptible to corrosion in the presence of chloride anions which takes place in a strongly exothermic reaction even at the OCP and proceeds despite the formation of corrosion products [43,46,48].

The “chemicals-free” intrinsic self-healing of PPM coating is supposed to be triggered by the exothermicity of the aluminum corrosion reaction [46] combined with the thermoplastic behavior and the relatively low temperature of the glass transition of the PPM copolymer that make possible “fluxing” of PPM copolymer in free spaces. As a further indirect confirmation of the healing process, the presence of an intact and continuous layer of the coating even on the site object of the confined corrosion events (bright areas under visible light) was confirmed by detecting the blue/violet emission of PPM copolymer under 386 nm light (Figure 6). Thus, despite co-PPM 9% possessing a T_g_ about 30 °C higher with respect to its analogue functionalized with octyloxy chains [10,11], it is still able to exploit the heat released during the corrosion reaction to trigger the self-healing.

## 3. Materials and Methods

### 3.1. Reagents and Solvents

Benzyl chloride stabilized by propylene oxide (99%), bismuth (III) trifluoromethansulfunate (98%), thionyl chloride (99%), and chloroform (99.8%, amylene stabilized) were purchased from Sigma Aldrich (Buchs, Switzerland), benzyltriethoxysilane (96%) from Fluorochem (Hadfield, UK), 4-methoxybenzylchloride (99%) from Apollo Scientific Ltd. (Whitefield, UK) and methanol (98%) from Merck (Darmstadt, Germany). Aluminum alloy AA2024 panels 4.5 cm in length, 3 cm in width and 5 mm in thickness (4.3–4.5% copper, 1.3–1.5% magnesium, 0.5–0.6% manganese, and less than 0.5% of other elements) were provided by Aviometal s.p.a. (Varese, Italy).

### 3.2. Side-Chain Engineering, Polymer Characterization, and Coating Preparation

The functionalization of the PPM polymeric backbone with different fractions of methoxy side chains was achieved by copolymerization of benzyl chloride and 4-methoxy benzyl chloride, varying the molar ratio of 4-methoxy benzyl chloride from 5% mol/mol to 10% mol/mol (see Synthetic procedure S1 section, in the Supporting Information file). The copolymerization was carried out by using Bi(OSO_2_CF_3_)_3_ as a catalyst and according to the procedure reported previously [10,11,43]. Copolymers containing, respectively, 4.6% and 9.0% (mol/mol) of methoxy side chains were synthesized, as established by ^1^H-NMR spectra. ^1^H-NMR (300 MHz, CDCl_3_, d): 3.16 (br, 0.136 H or 0.27H, respectively, -OCH_3_), 3.72 (br, 2H, -CH_2_-Ar), 6.95 (br, 4H, Ar).

The molecular weight of the copolymers was evaluated by gel permeation chromatography (GPC) using a Viscotek GPC system (Malvern, Worcs, UK) equipped with three columns (two PLGel Mix-C and on ViscoGEL GMHHRN 18055) and operated with tetrahydrofuran as mobile phase. The thermal properties were assessed by a TGA/DSC 3+ STAR^e^ system (Mettler-Toledo, Schwerzenbach, Switzerland) with sample crucibles of 70 mL fixed gap (0.7 mm).

Optical spectroscopy investigation was performed on diluted solution (0.5 wt.%, in THF) of the copolymer containing 9.0% (mol/mol) of methoxy side chains. Geometry Jasco FP-8500 fluorometer and Jasco V-660 were employed to study the photoluminescence emission and the UV–Vis absorption features of the material, respectively.

The coatings were prepared according to a procedure previously reported [10,11]. In order to remove imperfections and debris from the surfaces of AA2024, alloy panels were polished several times with abrasive paper (300, 800, 1200, 4000 grit) and washed in an ultrasonic bath (Banderlin, Berlin, Germany) by immersion in ethanol. Benzyltriethoxysilane was deposited on the polished aluminum surfaces by spin coating (3500 rpm, 30 s) and heated up to 100 °C for 1 min in order to allow the condensation of the primer [10,11]. Then, 150 mg of functionalized PPM was processed to 30 µm thick protective coatings with a hot press (120 °C, 8 kPa) [10,11,12].

### 3.3. Evaluation of Corrosion Protection and Intrinsic Self-Healing via Electrochemical Techniques

The corrosion protection abilities of the copolymers were investigated by exposing a PPM-coated aluminum specimen (working electrode), with an exposed area of 0.95 cm^2^, to a naturally aerated aqueous solution containing 3.5% w/v NaCl (0.6 mol L^−1^) in a single compartment cell, at room temperature. A saturated calomel electrode (SCE) was used as the reference electrode inserted in a Luggin capillary, its end being placed as close as possible to the exposed circular spot of the working electrode. Eventually, a platinum coil served as the counter electrode.

Coated AA2024 plates were tested according to three different protocols (Figure 7), as followed:(1)Potentiodynamic anodic polarization technique. After an equilibration period of 10 min at the open circuit potential, the coated electrode was polarized by scanning the potential, from the OCP, in the anodic direction with a scan rate of 10 mV min^−1^.(2)Anodic aging test. It is a cyclic procedure that includes three steps: (i) anodic polarization at 0 V vs. SCE for 10 min (to force corrosion reactions of the substrate in correspondence of defects or porosities of the coating), (ii) relaxation period at resting potential for 20 min, and (iii) electrochemical impedance spectroscopy (EIS) measurements at OCP to evaluate the new established condition.(3)Coating damaging. The time evolution of artificially damaged coated samples was monitored over 60 days by alternating OCP and EIS measurements. The coating was damaged creating a hole with a stainless-steel needle (diameter = 0.52 mm). The success of the damage was confirmed by verifying the electric continuity between the needle and the AA2024 substrate.

**Figure 7 ijms-23-16103-f007:**
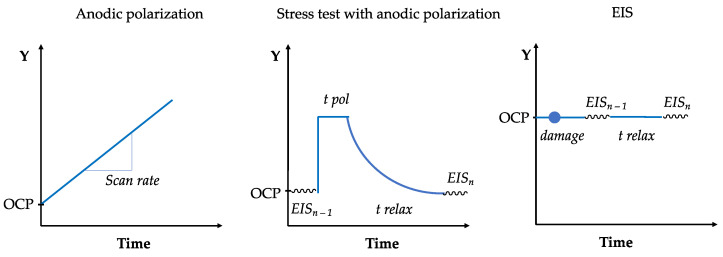
Schemes of the electrochemical tests performed in this work: potentiodynamic anodic polarization (**left**), anodic aging stress test (**middle**), and testing of artificially damaged samples (**right**). Y-axes report the electric potential (vs. the operating reference electrode), OCP is the open circuit potential, *t*_pol_ is the duration of the anodic polarization, *t*_relax_ is the relaxation time at the OCP. Damage corresponds to the time when the artificial scratch was done on the coating. Times are not scaled properly.

EIS analyses were performed applying to the coated electrode a potential sinusoidally oscillating around the OCP value (10 mV amplitude) in a range of frequency from 100 kHz to 0.1 or 0.01 Hz. The data were collected by a Biologic-VMP3 electrochemical workstation. The EIS data were analyzed by modelling experimental spectra by EC-Lab software (Bio Logic Science Instrument) drawing equivalent electric circuits able to reproduce the empirical data. Each circuit was composed by a resistance and different time constant elements (TCE) constituted by a parallel connection of a resistor and a capacitor (Figure 8). The circuit comprise always an electrolyte resistance (R_s_). A first-time constant element accounts for the pore resistance (R_po_), representing the measure of coating’s porosity and its degree of deterioration, and the coating capacitance (C_c_) which is proportional to water permeation inside the matrix. The second TCE is constituted by the polarization resistance (R_p_), representing the resistance at metal|electrolyte interface (inversely proportional to the kinetics of the corrosion process, at fix real area), and the double layer capacitance (C_dl_), measure of the delaminated area [49,50].

After electrochemical characterizations, the coating surfaces were exposed to UV light (395 nm) and investigated by means of an optical microscope (Wild Photomakroskop M400, Heerbrugg, Switzerland) equipped with a digital camera.

## 4. Conclusions

Based on the side-chain engineering approach, a new class of poly(phenylene methylene)-based copolymers was designed to improve the processability issues of the PPM homopolymer by exploiting the concept of the internal plasticizer. PPM-based copolymers containing methoxy side chains were synthesized with two different molar ratios of –OMe units per phenylene methylene units, 4.6% (co-PPM 4.6%) and 9.0% (co-PPM 9%). The polymers are thermally stable (decomposition > 400 °C) with a glass transition temperature (T_g_) of about 65 °C, well above that of the octyloxy PPM-copolymer described so far [10,12], that can extend the range of application of such class of thermoplastic coatings. While T_g_ is close to that of PPM homopolymer, the methoxy copolymers do not suffer from the brittleness that prevents obtaining crack-free films of PPM homopolymer [12]. Hence, through hot-pressing of copolymers powder, AA2024 plates were coated with 30 µm thick layers of co-PPM 4.6% and co-PPM 9% for corrosion protection purpose.

While co-PPM 4.6% coatings failed the potentiodynamic anodic polarization tests, resulting in the partial detachment of the coating from the substrate, co-PPM 9% coatings revealed a stable anodic current density in the range of pA cm^−2^ up to a +3 V overpotential, orders of magnitude lower than that of other PPM derivatives previously investigated [10,12]. The short-lasting metastable pit corrosion events detected by the chronoamperograms did not affect the overall integrity of the film. According to the general accepted rule, the improved protective performance of –OMe derivatives over the –OOct ones is attributed to the stronger intermolecular interactions between the polymer chains (i.e., better physical barrier effect) because of the shorter pendants.

Electrochemical testing under an accelerated aging condition and on artificially damaged coatings, with the support of the optical microscopy, indirectly proved the occurrence of a “chemicals-free” self-healing for co-PPM 9% coatings, a process that is triggered by the corrosion process itself. While the healed coating does not completely recover the original barrier property, it can mitigate and delay corrosion even under accelerated stress conditions.

In conclusion, the side chain engineering approach is able to produce processable PPM-based coatings endowed with self-healing feature still maintaining the relatively high glass transition temperature of the brittle PPM homopolymer. Remarkably, the discovery would open the way to the obtainment of new PPM-based anticorrosion coatings designed for higher temperature applications.

## Figures and Tables

**Figure 1 ijms-23-16103-f001:**
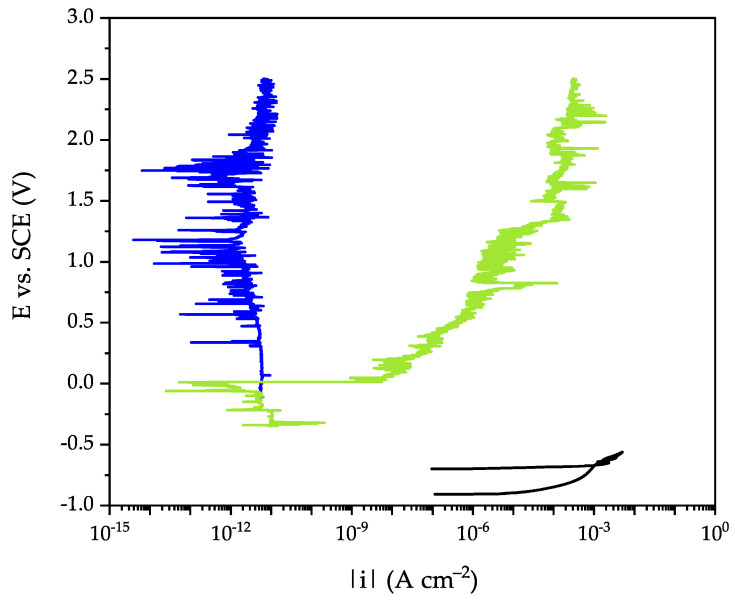
Representative potentiodynamic anodic polarization curves obtained for naked AA2024 (black; cyclic polarization) and for AA2024 coated by co-PPM 4.6% (green) and co-PPM 9.0% (blue) in naturally aerated 3.5% w/v NaCl solution. Potential scan rate: 10 mV/min.

**Figure 2 ijms-23-16103-f002:**
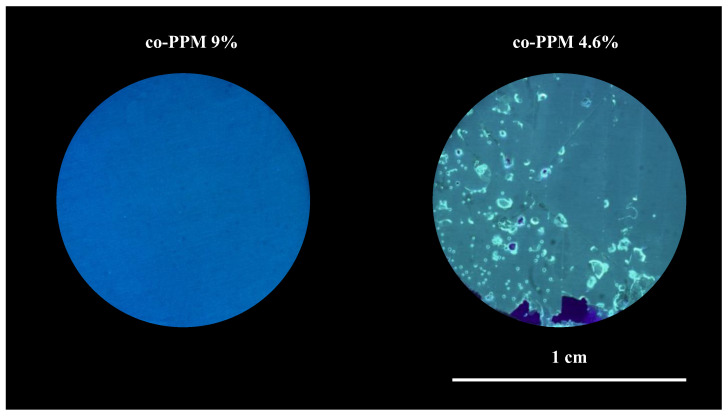
Optical microscope images (under 386 nm UV light) of the surface of 30 μm PPM copolymer coatings after the anodic potentiodynamic polarization. From (**left**) to (**right**): co-PPM 9% and co-PPM 4.6%.

**Figure 3 ijms-23-16103-f003:**
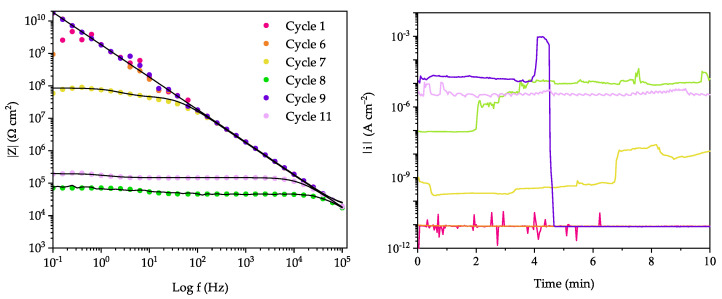
Evolution of the Bode modulus spectra at OCP (**left**; fitted spectra as black lines) recorded after each potentiostatic anodic polarization at 0 V vs. SCE (**right**) of a co-PPM 9% coating. Same colors for the two plots: magenta 1st cycle, orange 6th, yellow 7th, green 8th, purple 9th, and pink 11th.

**Figure 5 ijms-23-16103-f005:**
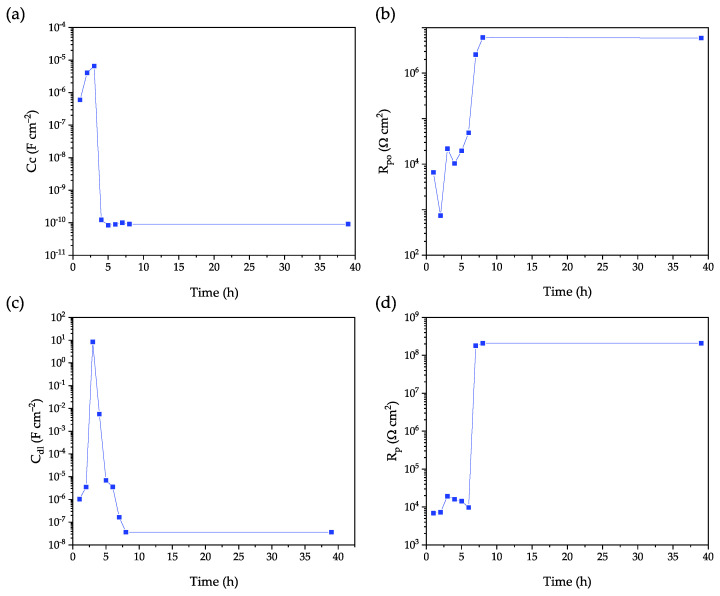
Evolution of the fitting parameters of the co-PPM 9% coating after the artificial damaging: coating capacitance, C_c_ (**a**); pore resistance, R_po_ (**b**); double layer capacitance, C_dl_ (**c**), and polarization resistance, R_p_ (**d**).

**Figure 6 ijms-23-16103-f006:**
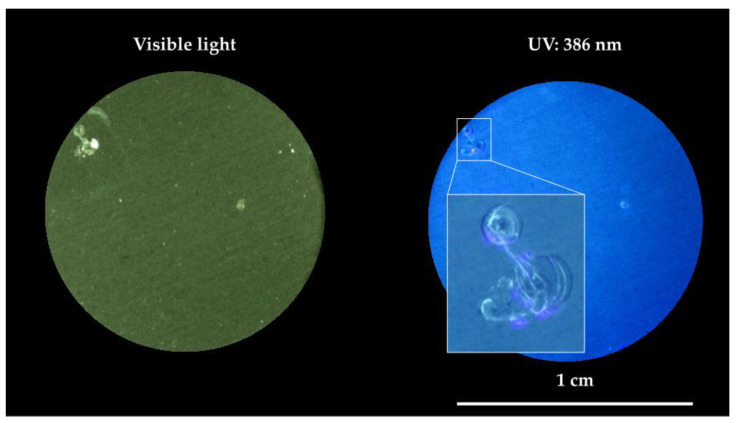
Optical microscope images of the surface of damaged co-PPM 9% coatings after 60 days of exposure to NaCl water solution.

**Figure 8 ijms-23-16103-f008:**
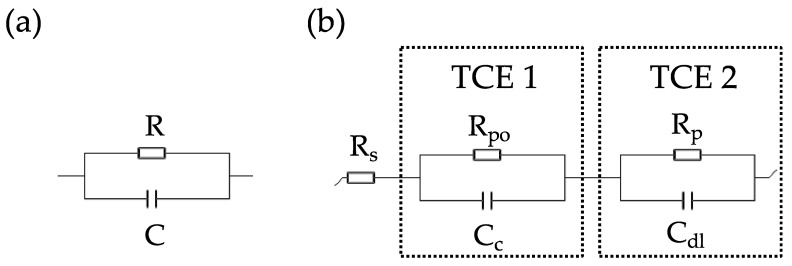
(**a**) A time constant element (TCE) composed by a resistor R with a capacitor C in parallel. (**b**) A two-time constant equivalent circuit, commonly used in this study: the electrolyte resistance is in series to a first TCE that models the polymer|electrolyte interface, and to a second TCE for the metal|solution interface.

## Data Availability

The data that support the findings of this study are available from the corresponding author upon reasonable request.

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
