# Peer review of "Improving the Corrosion Protection of Poly(phenylene methylene) Coatings by Side Chain Engineering: The Case of Methoxy-Substituted Copolymers"

_ijms, 2022, doi:10.3390/ijms232416103_

Round 1

Reviewer 1 Report

The article is interesting, but it needs to be structured by the data presented.

1. Introduction

- The authors have very well substantiated the relevance of this research, but they have not formulated the purpose. I recommend that the authors rework the last paragraph of the introduction. It is not necessary to write about the methods of research in the introduction and about the materials used. There is a section 4 for that. Please write clearly at the end of the introduction about the purpose of your research. 

2. Results and discussions

- Subsections 2.2, 2.4 - pay attention to the authors that the first mention of the figure in the text of the Manuscript comes first, and then the mentioned figure appears. Please make changes accordingly for figures 1, 4. 

- The preceding comment also applies to figures 2,3,5 and 7. I recommend the authors to place these figures at least at the end of the paragraph after the first mention in the text. For example, figure 2 is first mentioned in line 146 - place figure 2 at the end of this paragraph. The first mention of Figure 3 appears on line 192 - place Figure 3 at the end of this paragraph. For Figure 5, the first mention occurs on line 264 - place Figure 5 at the end of this paragraph. Figure 7 is first mentioned in the text on line 368 - place Figure 7 at the end of this paragraph, i.e., on line 384.

- It is necessary to give a reference in the text to figures 6 and 8 taking into account the previous remarks (i.e., first there is text with a link to the figure, and after this link, the figure appears).

3. Materials and Methods

- In the methodology, the authors mention that the TGA and DSC methods were used to study the synthesized materials. However, these are not present in the submitted materials. In that case, there are several ways: either they should be added, or this information should be deleted, or reference should be made to the article where these data were presented earlier.

- Also, pay attention to the fact that there are optical photographs in the article, although the authors do not write about this method of research in the methodology. Information about this method should be added to section 3.

- In addition, according to the images obtained using an optical microscope, we can conclude that the authors conducted exposure of the samples in a 3.5% NaCl solution for 60 days (Figure 6). However, there is not a word about this in the methodology. It follows that this information should be added to section 3. 

  • 4. Other Comments

- I recommend that the authors eliminate the supplementary file and move this information to the appropriate sections of the Manuscript. By adding this data to the Manuscript, the article would only benefit from it. In any case, the data presented in the supplementary file needs to be worked on. Please make Figure S1 and Figure S2 more readable. The X-axis is not its name, please correct it.

- In the text of the manuscript sometimes the words merge, please make the appropriate changes: compounds[21], degradation[22], synthesized[10,23], plasticizers[10,24,25], shocks[10,11], aluminum[11,26], temperature[10],  coating[24,27–29], temperatures[30–32],  coatings[24,33–36], decreases[28,35,37], coatings[28,29,38–40], reported[10,18,42], ppm[42], derivatives[10,18], interface. [18][28,29,35,37–40]Consequently, events[11,45],  in literature[18], coatings[11]. [2– 5,44]Thus, far[10,12].

- Please use the template for the reference list. At the moment not all the references are properly formatted, please make the necessary changes.

Author Response

Reviewer 1

  1. Introduction

- The authors have very well substantiated the relevance of this research, but they have not formulated the purpose. I recommend that the authors rework the last paragraph of the introduction. It is not necessary to write about the methods of research in the introduction and about the materials used. There is a section 4 for that. Please write clearly at the end of the introduction about the purpose of your research. 

Answer: We thank the Reviewer for his/her valid observation. We have changed the last paragraph, avoiding describing the methodologies but focusing on the purpose of the present work.

  1. Results and discussions

- Subsections 2.2, 2.4 - pay attention to the authors that the first mention of the figure in the text of the Manuscript comes first, and then the mentioned figure appears. Please make changes accordingly for figures 1, 4. 

- The preceding comment also applies to figures 2,3,5 and 7. I recommend the authors to place these figures at least at the end of the paragraph after the first mention in the text. For example, figure 2 is first mentioned in line 146 - place figure 2 at the end of this paragraph. The first mention of Figure 3 appears on line 192 - place Figure 3 at the end of this paragraph. For Figure 5, the first mention occurs on line 264 - place Figure 5 at the end of this paragraph. Figure 7 is first mentioned in the text on line 368 - place Figure 7 at the end of this paragraph, i.e., on line 384.

Answer: Thank for having noticed these incongruences. We have now amended them.

- It is necessary to give a reference in the text to figures 6 and 8 taking into account the previous remarks (i.e., first there is text with a link to the figure, and after this link, the figure appears).

Answer: We have properly solved these two missing references. Thank for having noticed them.

  1. Materials and Methods

- In the methodology, the authors mention that the TGA and DSC methods were used to study the synthesized materials. However, these are not present in the submitted materials. In that case, there are several ways: either they should be added, or this information should be deleted, or reference should be made to the article where these data were presented earlier.

Answer: Thermal analyses (TGA and DSC) were briefly commented in the original manuscript in the last paragraph of section 2.1. The thermal stability of the novel PPM copolymer was evaluated by TGA that reveals absence of any loss of mass up to 400 °C. DSC shows a phase transition (glass one) at 65°C, neatly higher than that of the octyloxy derivative and quite comparable to that of the homopolymer PPM.

The related discussion has been now further improved, to better underline the achievement of this important goal. For sake of completeness, we have also added TGA and DSC traces in the Supporting Information file (Figure S3).

- Also, pay attention to the fact that there are optical photographs in the article, although the authors do not write about this method of research in the methodology. Information about this method should be added to section 3.

Answer: We thank the Reviewer also for this comment. We apologize for the oversight. We have amended by adding the proper description at the end of Section 3.3.

- In addition, according to the images obtained using an optical microscope, we can conclude that the authors conducted exposure of the samples in a 3.5% NaCl solution for 60 days (Figure 6). However, there is not a word about this in the methodology. It follows that this information should be added to section 3. 

Answer: Once again, the careful revision of the Reviewer helps us to improve the clarity of the paper. Actually, we described the mentioned protocol in the 3.3 section (point 3 of the list), but we agree with the Reviewer that an additional comment/explanation is duly. For this reason, we have better discussed this protocol in the Result and Discussion session (among page 6 and 7, referring to numbers of right top of pages).

  1. Other Comments

- I recommend that the authors eliminate the supplementary file and move this information to the appropriate sections of the Manuscript. By adding this data to the Manuscript, the article would only benefit from it. In any case, the data presented in the supplementary file needs to be worked on. Please make Figure S1 and Figure S2 more readable. The X-axis is not its name, please correct it.

Answer: We have properly named the X-axes of Figure S1 and S2. We prefer to leave the Supporting Information file, as it has been now further enriched. We think that moving all the material in the main text is too much. We hope the Reviewer can understand our point of view.

- In the text of the manuscript sometimes the words merge, please make the appropriate changes: compounds[21], degradation[22], synthesized[10,23], plasticizers[10,24,25], shocks[10,11], aluminum[11,26], temperature[10],  coating[24,27–29], temperatures[30–32],  coatings[24,33–36], decreases[28,35,37], coatings[28,29,38–40], reported[10,18,42], ppm[42], derivatives[10,18], interface. [18][28,29,35,37–40]Consequently, events[11,45],  in literature[18], coatings[11]. [2– 5,44]Thus, far[10,12].

Answer: Thank you very much the very careful list of typos above provided. We have amended all of them.

- Please use the template for the reference list. At the moment not all the references are properly formatted, please make the necessary changes.

Answer: reference list has been revised.

Reviewer 2 Report

Some spectroscopic method and DFT studies are highly suggested to be added for further explanation.

A standard deviation, χ2 of the numerical values of impedance parameters and the relative error of each element?

Author Response

Reviewer 2

Some spectroscopic method and DFT studies are highly suggested to be added for further explanation.

Answer: We thank the Reviewer for his/her suggestion. We have added a discussion on the optical spectroscopy characterization carried out for the novel methoxy-bearing PPM copolymers. Please, have a look at the end of section 2.1 in the main text and at the related Figure S4 in the Supporting Information file.

No DFT studies were performed in this manuscript for two main reasons. Firstly, the results from the aforementioned spectroscopic characterization clearly point out that the introduction of the methoxy sidechains in the polymer backbone (up to 9% mol/mol) does not seem to modify the energetics of the frontier molecular orbitals. Secondly, we have not the suitable knowledge and software to run DFT calculations to study our system. We hope the Reviewer can understand our motivations.

A standard deviation, χ2 of the numerical values of impedance parameters and the relative error of each element?

Answer: The statistical relevance of the data provided in the manuscript concerning the EIS study seems to be poor, but there is an explanation. The reason is related with the intrinsic “random nature” of the events we tried to “snap/freeze” in the Figures 3-5. As a consequence, the occurring always at a fixed time of a specific event (e.g., self-healing, significant water uptake) is out of the control of the operator. This is the reason why we cannot provide average and related standard deviations for i) the EIS spectra (Figures 3, 4a in the main text; Figures S6 and S7 in the Supporting Information file) and ii) the values of the key parameters selected to depict the evolution in time of the coated system (Figures 4b and 5). In trying to reporting the average values, and the related error bars (in spectra and plots), we would obtain something that is significantly affected by the random occurring of the events responsible for the shape of the spectra. Hence, many artefacts would be generated negatively affecting the trend and the interpretation of the data.

Nonetheless, we want to underline that each measurement was repeated at least in triplicate (up to 5-6 times) under the same conditions (and for each single protocol shown in the manuscript). Therefore, graphs reproduce a representative trend, without any claim to report values valid in absolute terms. What matters is the variation over time of the parameters (and of the related impedance spectra) as by this trend we can identify events that occur at microscopic level.

Concerning the fitting procedure. For a given spectrum (part of a series of spectra recorded at subsequent times), the relative error of the elements of the EECs used are no more than 60-70% in almost all cases. This relatively high value is mainly due to the “noised points” recorded for these samples, as a result of the very high insulating capability of the coating (with |Z| ranging between 107 to 109 Ω as average, but reaching values even up to tens of GΩ). Once again, values of the circuit elements are not to be taken in absolute value, but as a whole to track their variation over time. In this view, considering the very wide range within which the values of the circuit elements span (at least three order of magnitudes), even relative errors up to many hundreds percent do not substantially modify the shape of the plots in Figure 5.

Round 2

Reviewer 1 Report

Dear authors, I see that you have made appropriate changes, thank you.

I pay attention that in the submitted version of the manuscript from page 6 to 16, further page 16 lines 767, 769, 773 - appears this strange phrase: ''Error! Reference source not found''. I suppose that this is some kind of technical error, it must be corrected. That's why I refer the article to minor revision.
Nevertheless, I recommend that this article be published.